# Eosinophilic Esophagitis in Esophageal Atresia: Is It Really a New Disease?

**DOI:** 10.3390/children9071032

**Published:** 2022-07-11

**Authors:** Camilla Pagliara, Elisa Zambaiti, Luca M. Antoniello, Piergiorgio Gamba

**Affiliations:** Department of Pediatric Surgery, University Hospital of Padova, 35100 Padova, Italy; camilla.pagliara24@gmail.com (C.P.); lucamaria.antoniello@aopd.veneto.it (L.M.A.); piergiorgio.gamba@unipd.it (P.G.)

**Keywords:** esophageal atresia, eosinophilic esophagitis, dysphagia

## Abstract

Eosinophilic esophagitis (EoE) is a chronic, immune-mediated esophageal disease. Symptoms are related to mucosal eosinophilic-predominant inflammation that leads to esophageal dysfunction. Recent data suggest that esophageal atresia (EA) patients may have an increased incidence of EoE compared to the general population. As EoE symptoms may be confused with EA-related symptoms, they may significantly worsen morbidity in this specific group of patients. We investigated specific characteristics of patients with AE and EoE compared to those with EoE only. We conducted an observational retrospective monocentric study including all patients diagnosed with EoE from 1 January 2010 to 31 December 2021. For each patient, demographic, clinical and histopathological data were collected and then compared between the two cohorts (EA-EoE vs. EoE only). During the study period, 62 patients were included: 17 children were in the follow-up because of EA (18.1% of 94 EA patients screened in that period), while the other 45 presented EoE only. The demographic and clinical features of EA-EoE patients demonstrate a lower prevalence of allergic subjects (23.5% vs. 80%, *p* < 0.05), a lower age of presentation (3.1 vs. 12.2 years, *p* < 0.05), non-specific symptoms and a higher resolution rate with PPI therapy (64.7% vs. 17.8%, *p* < 0.05) compared to EoE-only patients. Our data confirm that EA patients are at high risk for developing EoE. As symptoms may overlap with the EA spectrum, early recognition of EoE may prevent patients from receiving unnecessary invasive therapeutic interventions and from developing complications from untreated EoE.

## 1. Introduction

Esophageal atresia (EA) is a rare congenital disease with an incidence of 1 in 2500 live births [1]. Although improvements in surgical techniques have resulted in increased native-organ survival and overall amelioration of outcomes, patients with EA manifest significant life-long gastrointestinal morbidity [2]. Post-operative anastomotic stricture is a common early complication, and it may require frequent endoscopic dilatation. Moreover, high incidences of feeding difficulties and failure to thrive have been reported, especially in younger patients. Finally, the vast majority of EA patients have a persistent degree of esophageal dysmotility that may predispose them to GERD [3].

Eosinophilic esophagitis (EoE) is defined as a chronic, immune-mediated esophageal disease induced by an extreme immunological reaction to food- and environment-related allergens [4]. EoE is considered a rare disease, with a reported prevalence of 1 to 5 per 10.000 in the general pediatric population; however, as we are currently gaining increasing insight into the disease pathogenesis and its recognition, EoE diagnosis and its overall prevalence are increasing [5]. In patients who develop EoE, the T-helper cell 2–type immune response plays a crucial role in specific interleukins’ activation and in eosinophil chemotaxis factor secretion, which are responsible for the differentiation, maturation and recruitment of eosinophils in the esophageal mucosa [6]. The eosinophilic-predominant inflammation localized in the esophageal mucosa ultimately leads to esophageal dysfunction with the development of symptoms [7]. Clinical manifestations vary depending on the age: in adolescents, dysphagia with food impaction and chest pain or heartburn is predominant, while in childhood, the diagnosis of EoE can be difficult due to an overlap of symptoms with gastroesophageal reflux disease (GERD), such as failure to thrive and food intolerance [8].

In this context, despite recent data suggesting that EA patients may have an increased incidence of EoE compared to the general population [9], the concomitant presence of EoE and EA may be difficult to suspect, as EoE partially overlaps with EA’s common gastrointestinal manifestations or GERD symptoms [8]. The aim of the present study is to report on the demographic and clinical characteristics of EoE in patients treated for EA compared to those of the EoE-only population.

## 2. Materials and Methods

An observational retrospective cohort description was conducted in all patients diagnosed with EoE from 1 January 2010 to 31 December 2021 in a single tertiary center. We included all patients diagnosed with EoE from the general population at our institution or elsewhere. Demographic, clinical, histopathological and therapeutic findings of patients with EA-EoE were compared with patients with EoE only.

For patients with both EA and EoE, the diagnosis was generally made during protocol follow-up for EA, which constituted the following. The institutional post-operative follow-up protocol includes administration of proton pump inhibitor (PPI) therapy for at least the first 6–12 months of life to all patients. At this point, endoscopy with biopsies and positioning of 24 h pH-metry is performed in order to identify precocious patients with GERD or other types of esophagitis. After that, depending on clinical and endoscopic trends, PPI may be suspended, and outpatient checks are scheduled. Further routine endoscopies with biopsies and 24 h pH-metry in asymptomatic patients are planned at 5, 10 and 15-year-old time points, according to ESPGHAN and ERNICA guidelines [3,10]. In the case of symptomatic patients (i.e., dysphagia, feeding intolerance, regurgitation and vomiting, food impaction, etc.), endoscopy with biopsies is performed whenever required in order to exclude anastomotic stricture and peptic or eosinophilic esophagitis; in the case of a negative endoscopy, 24 h pH-metry is also recommended.

EoE was diagnosed from both clinical and histological features according to the criteria of Dellon et al. [11]. Symptoms such as food intolerance, dysphagia, heartburn, vomiting and episodes of food impaction were considered suggestive of EoE. Esophageal endoscopy with multiple biopsies was performed in order to confirm the diagnosis: linear furrows, white plaques and esophageal rings were recognized as typical features of EoE, but the presence of more than 15 eosinophils in at least one high-power microscopy field (HPF) established the diagnosis.

For each patient, demographic, clinical and histopathological data were collected, focusing on the disease history and outcome of EA-EoE. The type of EA was classified according to Gross classification (A-E classification) [12]. Long-gap EA was defined when the anatomical distance between the proximal and distal esophageal segments corresponded to 3 or more vertebral bodies [10].

Once the diagnosis was established, all patients were followed up in a multidisciplinary group with a dedicated pediatric gastroenterologist and pediatric surgeons. According to the recommendation mentioned above, children were first treated with PPI medication. Initial PPI doses of 1 mg/kg per dose were employed twice daily. Patients without clinical and histological responses to PPI were administered swallowed topical corticosteroids: children aged 1–10 years old were treated with fluticasone propionate 110 µg/puff inhaler, 2 puffs twice per day, or oral viscous budesonide 0.5 mg; patients older than 11 were treated with fluticasone 220 µg/puff inhaler, 2 puffs twice per day, or budesonide 1 mg. Endoscopic reassessment was performed in order to confirm histological remission.

Categorical variables are reported as proportions, while continuous variables are reported as medians with their ranges (IQR). Outcomes between groups were compared using the Χ² test, Fischer’s exact test and Mann–Whitney U test as appropriate. Statistical analysis was conducted, and results were displayed using GraphPad Prism 8.3.0. A *p*-value < 0.05 was considered statistically significant.

This retrospective study involving human participants was in accordance with the Helsinki Declaration. Ethical approval was waived by the local Ethics Committee as per the center protocol in view of the retrospective nature of the study, and all of the procedures being performed were part of routine care. Every participant gave informed consent to participate in the research. This research received no external funding.

## 3. Results

A total of 62 patients with EoE were included in the study. Seventeen children (18.1% of the ninety-four EA patients screened in that period) with EA-EoE were identified. Consequently, 45 children with only EoE were diagnosed.

Among the EA-EoE population, median weight and gestational week at birth were respectively 2350 g (IQR 2130—2600 g) and 36.5 gestational weeks (IQR 35–39 GW). EA with distal tracheo-esophageal fistula (type C according to Gross classification) was the predominant subtype (14; 82.4%), and in 12 cases (70.5%), associated malformations were reported, with a high prevalence of skeletal anomalies (6; 35.3%), cardiac defects (6; 35.3%) and anorectal malformation (5; 29.5%). A common skeletal anomaly was vertebral schisis (3; 17.6% of total); the most representative cardiac defect was a ventricular septal defect (5; 29.5% of total). In three patients (17.6%), a syndromic association was detected. Seven (41.2%) long-gap atresia cases were described, while in the other cases, primary anastomosis could be achieved during the first operation a few days after birth. Thoracotomy was the technique of choice in 13 patients (76.4%). In three cases (17.6%), traction sutures were applied in order to ensure the elongation of the two esophageal segments. In four patients (23.6%), the surgeon described an esophageal anastomosis under tension, and in one case (5.8%), post-operative anastomotic dehiscence occurred. Twelve children (70.5%) were treated for post-operative anastomotic stricture with serial endoscopic dilatation during the first year of life. The median number of endoscopic dilatations was 3 (IQR 2–5 dilatations). Six patients underwent gastrostomy placements (35.3%), all before the diagnosis of EoE.

A comparison of clinical characteristics between the EA-EoE group and EoE patients from the general population is reported in Table 1, Table 2 and Table 3.

The male-to-female prevalence ratio was quite similar between the two groups (1.9 vs. 3.1 for EA-EoE and EoE groups, respectively, *p* = 0.40). An allergic history was typical among the EoE population, while it was uncommon in EA-EoE patients (*p* < 0.001). Out of 17 EA-EoE patients, only 4 (23.6%) had a food and/or environmental allergy, confirmed by skin-prick tests and/or radioallergosorbent tests (RAST). A history of reactive airway disease was reported in nine children with EA (53.0%), including allergic rhinoconjunctivitis (3; 17.6%) and asthma (3; 17.6%) as the most representative.

A history of GERD was observed in 11 children with EA-EoE (64.7%). The diagnosis was confirmed with standard radiological tests (an esophagogram and/or pH-metry). Due to dysphagia and frequent episodes of gastroesophageal reflux, eight children (47.0%) underwent an antireflux procedure prior to the diagnosis of EoE; all of them had pH-metry indicative of GERD and esophageal biopsies negative for EoE when antireflux surgery was performed. Among the EoE group, only 20 children were tested for GERD, and in about half of them (11, 55.0%), the diagnosis was confirmed (*p* = 0.54).

The median age at EoE diagnosis was 3.1 years and 12.2 years in the EA-EoE and EoE populations, respectively (IQR 1.4–10.1 years vs. 7.9–13.2 years; *p* < 0.0001). Eleven children with EA-EoE (64.7%) received the diagnosis during one of their routine follow-up endoscopies, while in the remaining group of patients, a semi-urgent endoscopy was performed due to the rapid onset of suggestive symptoms, and in one case, the diagnosis was made because of an episode of food impaction requiring endoscopic removal. Conversely, among the EoE-only cohort, EoE was always diagnosed during an endoscopy performed because of suggestive symptoms or, in eight children (17.8%), during an acute episode of food impaction. Symptoms at the time of endoscopy are listed in Table 2.

Five patients (29.5%) in the EA-EoE group and eight children (17.8%) in the EoE-only group had a normal endoscopy (*p* = 0.31). Erythema and ulceration were present in the majority of the patients with EA-EoE, 41.2% and 23.6% of cases, respectively. Typical endoscopic signs of EoE, such as linear furrows or white plaques, were noted in a few patients, and none presented esophageal rings, while they were the most representative endoscopic findings in the EoE population (see Table 3).

Six patients had a stricture at baseline endoscopy at the time of diagnosis of EoE. Among the EA-EoE group, the location of the stricture was at the site of EA anastomosis in two cases out of three; only one required pneumatic dilatation at the time of diagnosis. In all three cases, after starting medical treatment for EoE, there was an improvement in the luminal diameter of the esophagus at the site of the previously documented stricture. Macroscopic findings and histological changes associated with EoE for both groups are summarized in Table 3.

Among patients with EA-EoE, 11 children had a response to PPI therapy (64.7%); of those, 6 were not on PPI treatment before the endoscopy, while 5 were on PPI at the time of biopsy: for the latter, the dose of PPI medication was increased, or the actual therapy was prolonged for another 3–6 months before repeating the endoscopy. The remaining six patients (35.3%) did not respond to PPIs and therefore required second-line treatment with swallowed topical corticosteroids. At the time of drafting the present paper, for the EA-EoE population, three patients were still receiving therapy (two with swallowed topical steroids and one with PPI), two had been lost to follow-up, and twelve were in remission. In contrast, only 1 out of 6 patients with EoE had a response to PPI therapy (8, 17.8%), while the others (37, 82.2%) required swallowed topical corticosteroids (*p* = 0.0003). The median time from the diagnosis of EoE to the last follow-up was 15.8 months (IQR 10.7–29.7 months) and 9.8 months (IQR 5.3–17.8 months) for EA-EoE and EoE groups, respectively (*p* = 0.05). Further, relapse was significantly more frequent for EoE-only patients (2, 11.7% vs. 21, 46.7%; *p* = 0.01).

## 4. Discussion

EoE is an emerging entity, the prevalence of which is currently 1 to 5 per 10,000 [5], despite increasing due to improved awareness and therefore more frequent diagnoses. In the last ten years, some studies have postulated a strict correlation between EoE and EA [9].

In the present study, we first evaluated the symptoms and clinical history of the EoE-EA population and confirmed that EA patients were at higher risk for developing EoE compared to the general pediatric population.

Many factors may explain these findings. The literature available on EoE prevalence in EA patients is contradictory: some studies identified a 20% prevalence [13,14,15], similar to the one that we report in the present study, while another recent cohort study detected a lower prevalence of less than 5% [16]. The main factor responsible for this discrepancy may be the different approaches to esophageal sampling: according to the ESPGHAN guidelines, endoscopy with biopsies is mandatory for the routine monitoring of GERD or any type of esophagitis in patients with EA; therefore, while Tambucci et al. chose a “selective” approach in symptomatic children, we routinely perform esophageal biopsies during all routine endoscopies of the protocol, i.e., at 6–12 months before stopping PPI therapy and, in the case of asymptomatic patients, at 5, 10 and 15 years [3]. Therefore, even if asymptomatic, EA patients represent a highly controlled population due to the well-known risk of esophageal morbidity due to anastomosis in the first years of life and later due to the risk of cancer progression in the following years (more than 100-fold higher than in the general population) [17]. This may be one of the reasons behind the higher incidence detected. In fact, among our cohort of EA-EoE patients, in the majority of cases, EoE was diagnosed at one of the routine follow-up endoscopies, during which they referred to tolerable generic gastrointestinal symptoms such as food intolerance, regurgitation and dysphagia. Thereafter, about half of children presented a macroscopically normal endoscopy, and EoE was diagnosed histologically.

Another explanation may be etiopathological, as in the EA population, impaired esophageal motility can lead to a wide spectrum of long-term respiratory and gastrointestinal morbidity [11]. With regard to the latter, esophageal dysmotility increases the risk of GERD; prolonged acid exposure time may impair the mucosal barrier function and make the mucosa permeable to small peptides that would be impermeable in normal conditions [18]. This may allow food allergens to enter the deep layer of the esophagus and induce eosinophilic inflammation. Furthermore, esophageal stasis due to congenital and/or postsurgical dysmotility increases the contact time between esophageal mucosa and food allergens, leading to chronic irritation, increased mucosal permeability and eosinophilic-predominant inflammation [19]. Finally, due to chronic GERD, EA patients tend to be treated with PPI therapy longer than normal children. Although PPI medication is considered part of medical treatment for EoE, some studies reported that long-term PPI therapy can neutralize gastric acid and prevent the break-down of food allergens, increasing the potential for sensitization to these agents [20]. Furthermore, some studies indicate a genetic association between EA and EoE through mutation in the FOX gene complex, which is involved in both causing congenital anomalies (i.e., EA) and promoting the gene transcription of eosinophil chemotaxis factor, which are increased in EoE [21].

As mentioned, among our cohort of EA/EoE patients, the most frequent symptoms were food intolerance and vomiting, both representative of refractory GERD (together with dysphagia and/or regurgitation). Furthermore, esophageal dysfunction due to eosinophilic infiltrates could exacerbate a concomitant GER disease [22]. In this setting, diagnosis of EoE can be very challenging, and patients have an increased risk of either developing complications from untreated EoE or being misdiagnosed as having a gastroesophageal reflux disease refractory to medical treatment and undergo unnecessary invasive interventions (i.e., antireflux procedures or nutritional stoma creation). Because of this high percentage of patients reporting reflux-like symptoms, which might constitute an overlap of comorbidities, we highlight the need to perform adequate biopsy mapping during endoscopic follow-ups, even when considering patients as candidates for surgical procedures, such as fundoplication or gastrostomy.

In the present study, we also aimed to compare demographic and clinical characteristics between EA/EoE children and a group of EoE patients from the general population. Our data suggest that children with EA-EoE present different demographic, clinical and response-to-therapy features compared to those of the EoE population.

First, it seems that EA patients may have an earlier onset of EoE than children from the general population; as postulated before, this particular group of patients undergoes routine surveillance endoscopy with adequate biopsy mapping, which might enable the early diagnosis of EoE, even when symptoms are still mild [16].

When a diagnosis of EoE is made in preschool-age children with EA, the most representative symptoms of the disease in the EA-EoE cohort seem to be vomiting and food intolerance [23], while older children from the general population with only EoE manifest the usual symptoms of EoE, such as dysphagia, food impaction and chest pain.

In the literature available, the prevalence of atopic comorbidities is also reported to be high in cohorts of EA-EoE patients [16,24]. However, although allergies and atopy were prevalent among our EoE cohort of patients, only four children with EA-EoE presented with an allergy history. Our data seem to suggest that the etiopathogenesis of EoE among the EA cohort could be related to esophageal dysmotility as an intrinsic part of their base disease instead of the exaggerated activation of the immune system, as supposed for the EoE-only general population.

Finally, these patients seem to respond better to PPI therapy than the general pediatric EoE population. In fact, treatment of EoE has evolved in the last few years, and nowadays, PPI-responsive esophageal eosinophilia is no longer considered an atypical manifestation of GERD but rather a subtype of EoE. From this perspective, PPI medications that restore mucosa integrity and prevent abnormal permeability and antigen penetration are better classified as a treatment for EoE than as a diagnostic exclusion criterion [11,25]. Moreover, PPIs also have anti-inflammatory properties by inhibiting the production of pro-inflammatory cytokines and adhesion molecules on the eosinophil cell surface [26].

The major limitation of our study is represented by its retrospective nature. In order to give strength to our study, further studies are required to confirm the pathogenetic hypothesis. Furthermore, prospective studies should devote more attention to defining updated diagnostic and therapeutic criteria specifically adapted for this subgroup of patients, as we already discussed that the presence of gastrointestinal symptoms in the EA population is misleading, as most of the patients consider generic gastrointestinal symptoms to be tolerable, whereas the general population would consider them anomalies.

## 5. Conclusions

Growing evidence indicates that EoE among patients with EA represents a different subtype of EoE with a different course of the disease, which requires the development of “modified” criteria for diagnosis and treatment strategies. Certainly, early recognition of EoE may prevent patients from receiving unnecessary invasive therapeutic interventions for presumed refractory GERD and from developing complications from untreated EoE, and, more broadly, the correct treatment may reduce esophageal morbidity in this specific group of patients.

## Figures and Tables

**Table 1 children-09-01032-t001:** Allergic features.

Type of Allergy	EA-EoE (17)	EoE (45)	*p*-Value
Allergy (general), n (%)	4 (23.6%)	36 (80.0%)	*<0.0001 °*
Food allergy, n (%)	1 (5.8%)	28 (62.2%)	*<0.0001 °*
Environmental allergy, n (%)	4 (23.6%)	31 (68.9%)	*0.0018 °*
Reactive airway disease, n (%)	9 (53.0%)	29 (64.4%)	0.40 *
Allergic rhinoconjunctivitis, n (%)	3 (17.6%)	22 (48.9%)	*0.04* °
Asthma, n (%)	3 (17.6%)	6 (13.3%)	0.69 °
Atopic dermatitis, n (%)	2 (11.7%)	13 (28.9%)	0.20 °

n = number; IQR = interquantile range; EoE = eosinophilic esophagitis; EA = esophageal atresia. * X^2^ Test; ° Fisher exact test, italic: statistically significative.

**Table 2 children-09-01032-t002:** Clinical features at diagnosis.

Clinics at Diagnosis	EA-EoE (17)	EoE (45)	*p*-Value
Age at diagnosis, years, median (IQR)	3.1 (1.4–10.1)	12.2 (7.9–13.2)	*<0.0001 #*
Follow-up routine endoscopy, n (%)	11 (64.7%)	0 (0%)	N/A
Acute esophageal bolus, n (%)	1 (5.8%)	8 (17.8%)	0.42 °
No symptoms reported, n (%)	3 (17.6%)	2 (4.4%)	0.12 °
Dysphagia, n (%)	8 (47.0%)	28 (62.2%)	0.38 *
Food intolerance, n (%)	9 (53.0%)	7 (15.6%)	*0.007* °
Food impaction, n (%)	9 (53.0%)	29 (76.3%)	0.41 *
Vomiting, n (%)	11 (64.7%)	17 (37.8%)	*0.05* *
Heartburn, n (%)	1 (5.8%)	8 (17.8%)	0.42 °
Chest pain, n (%)	2 (11.7%)	12 (26.7%)	0.31 °

n = number; IQR= interquantile range; EoE = eosinophilic esophagitis; EA = esophageal atresia. * X^2^ test; # Mann–Whitney Test; ° Fisher exact test, italic: statistically significative.

**Table 3 children-09-01032-t003:** Endoscopic and histologic signs at diagnosis.

Macroscopic Characteristics	EA-EoE (17)	EoE (45)	*p*-Value
Normal Endoscopy, n (%)	5 (29.5%)	8 (17.8%)	0.32 °
Stricture, n (%)	3 (17.6%)	3 (6.7%)	0.33 °
Anastomotic strictures, n (%)	2 (11.7%)	0 (0%)	N/A
Endoscopic dilatation, n (%)	1 (5.8%)	1 (2.2%)	0.47 °
Erythema, n (%)	7 (41.2%)	7 (15.6%)	*0.04 °*
Ulceration, n (%)	4 (23.6%)	5 (11.1%)	0.24 °
Linear furrows, n (%)	1 (5.8%)	23 (51.1%)	*0.001 °*
White plaques, n (%)	2 (11.7%)	20 (44.4%)	*0.01 °*
Esophageal rings, n (%)	0 (0%)	23 (51.1%)	*<0.0001 °*
**Microscopic Characteristics**			
Number of Eo, median (IQR)	18 (16–30)	35 (23.5–41)	*0.002 #*
Superficial Eo, n (%)	7 (41.2%)	33 (73.3%)	*0.02 **
Microabscesses, n (%)	3 (17.6%)	14 (31.1%)	0.35 °
Degranulated Eo, n (%)	15 (88.3%)	44 (74.6%)	0.18 *
Focal Eosinophilia, n (%)	2 (11.7%)	5 (11.1%)	>0.99 °
Intercellular edema, n (%)	6 (35.3%)	14 (31.1%)	0.76 *
Basal zone hyperplasia, n (%)	10 (58.8%)	34 (75.5%)	0.19 *
Papillary elongation, n (%)	4 (23.6%)	2 (4.4%)	*0.04 °*
Subepithelial hyalinization, n (%)	3 (17.6%)	4 (8.9%)	0.38 °
Lamina propria fibrosis, n (%)	6 (35.3%)	10 (22.2%)	0.29 *

n = number; IQR = interquantile range; EoE = eosinophilic esophagitis; EA = esophageal atresia; Eo = eosinophils. * X^2^ test; # Mann–Whitney Test; ° Fisher exact test, italic: statistically significative.

## Data Availability

All data are reported in this manuscript.

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
