# Peer review of "Eosinophilic Esophagitis in Esophageal Atresia: Is It Really a New Disease?"

_children, 2022, doi:10.3390/children9071032_

Round 1
Reviewer 1 Report
The present retrospective study provides data of EoE in EA patients in a single tertiary center. The authors declare an observational study design in a retrospective study with only one group! The topic is up to date as literature is rare (32 results in pubmed search for EoE and EA, 9 reviews). However, the article requires a literature update, as some citations are more than 20 years old and recent reviews are missing.
The introduction is confusing as initially EoE is presented followed by an overview of EA. It would be easier to follow in the other way round, as EoE is discussed as a consecutive disease of EA.
In the methods section shows methodological inaccuracies and missing controls. As mentioned above, a retrospective cohort description is not an observational study, an age- and sex matched control group is missing. The design is not appropriate to test the hypothesis, as the authors conclude an earlier onset of EoE and a better respond to PPI therapy. An adequate control group would be GERD patients without history of EA.
The results require improvement: use n=XX, XX% as standard, give Mean and SD or Median and IQR. The Gross classification uses A – E and not type 3. Use subgroups to present the data, the reader is confused of percentage of total (EoE or EA). Long gap was diagnosed in 7 patients, but primary repair could be achieved in almost all patients (definition!). Were biopsies taken in the patients with fundoplication prior to the anti-reflux surgery (117, no EoE?). Give numbers in the results sections and avoid “few, many, some…” What happened to the EoE patients, who were already treated with PPI? Has the dose been increased? At the time of drafting the present publication three patients were still on therapy: Please give us more details about these patients: Do you treat them for month with swallowed topical steroid? Please correct the total number (n=17): 12 were in remission, 1 was lost, 3 were still on therapy? The whole section 164 – 172) is listed twice (before Table 2)!
As stated above a higher risk for EoE, earlier onset and better respond to PPI therapy cannot be concluded, because there is no adequate control group.

Reviewer 2 Report
This is an interesting manuscript. However, performing an endoscopy even in asymptomatic patients seems controversial. The authors should give very good reasons for following that protocol. In addition, the authors should improve and/or clarify some aspects before considering this article for publication:
Do the authors always perform a 24h phmetry before the endoscopy? Do the authors perform an endoscopy even if the phmetry is normal?
I would recommend explaining a little bit more your medical treatment protocol in patients with eosinophilic esophagitis including drug dosages.
Please clarify how many patients had GERD plus eosinophilic esophagitis in your study.
The last paragraph of the "results" part is repeated, please correct that.
Round 2
Reviewer 1 Report
The authors have revised the manuscript and improved the scientific standard. However, some parts, especially the abstract, are still confusing and minor spelling mistakes should be improved:
13: all patients diagnosed with EoE and underwent an endoscopy with esophageal biopsies from 01.01.2010 to 31.12.2021.
Revise the abstract and explain the methods: EA-EoE group compared to EoE only
Give a structured overview: 94 patients with EA were included, 17 of them (18,1%) were diagnosed with EoE. 45 otherwise healthy patients with EoE only were diagnosed in the same time period.
17: bring the statistically significant result into the abstract regarding the age (p<0.0001), followed by the other findings between these two groups, especially significant findings (allergy)
109: Please give a better overview as mentioned above: total 94 (better use numbers) patients with EA were included, 17 of them (18,1%) were diagnosed with EoE. 45 otherwise healthy patients with EoE only were diagnosed in the same time period.
115: mention the Classification Gross type C
134 Out of 17 EA-EoA patients
158: This is confusing: In general population EoE is diagnosed… in eight children?
169: change to EA-EoE group and EoE only group
192: Please give information about the remaining 6 patients (remission? Follow-up?)

Reviewer 2 Report
Congratulations on this new version of the manuscript.I would suggest adding the incidence of malignization in the sentence "EA patients represent a highly-controlled population, due to the well-known risk of esophageal morbidity due to the anastomosis in the first years of life and later due to risk of cancer degeneration in the following years "
In addition, I would resume the discussion part, as it is too long and some information is reiterative.
